# “I Called When I Was at My Lowest”: Australian Men’s Experiences of Crisis Helplines

**DOI:** 10.3390/ijerph19159143

**Published:** 2022-07-27

**Authors:** Katherine Trail, Michael J. Wilson, Simon M. Rice, Tara Hunt, Jane Pirkis, Zac E. Seidler

**Affiliations:** 1Orygen, Parkville, VIC 3052, Australia; michael.wilson@orygen.org.au (M.J.W.); simon.rice@orygen.org.au (S.M.R.); zac.seidler@orygen.org.au (Z.E.S.); 2Centre for Youth Mental Health, The University of Melbourne, Melbourne, VIC 3010, Australia; 3Lifeline Research Foundation, Lifeline Australia, Sydney, NSW 2000, Australia; tara.hunt@lifeline.org.au; 4Centre for Mental Health, Melbourne School of Population and Global Health, University of Melbourne, Melbourne, VIC 3010, Australia; j.pirkis@unimelb.edu.au

**Keywords:** telephone crisis helpline, suicide prevention, crisis intervention, COVID-19, men’s health, help-seeking

## Abstract

**Background:** Helplines are an accessible form of support for people struggling with difficulties in their lives and are key services in suicide prevention and intervention. Men’s experiences of telephone helplines are not well understood, despite high male suicide rates. **Methods:** We conducted an online cross-sectional survey with N = 684 Australian men (aged 17–83 years, *M* = 50.13) using open- and closed-ended questions about their experiences of helplines during the COVID-19 pandemic. Descriptive statistics were analysed to investigate differences between men using and not using helplines. Qualitative responses were analysed using thematic analysis. **Results:** Within the sample, 100 men (14.6%) had used a helpline service. Men using helplines were more likely to be unemployed and in younger age brackets than those not using helplines. They were also more likely to report experiencing stressors related to COVID-19, including financial stress and job loss, perceived impact on mental health and relationship breakdown. Qualitative analysis indicated varied experiences of helplines, with men shedding light on how their interaction with a counsellor, the structure of services and their expectations of the service impacted their experience. **Conclusions:** Further in-depth qualitative enquiry in this space is required, with the objective of understanding how helpline services may seek to better engage with male callers.

## 1. Introduction

Men in Australia are 3–4 times more likely to die by suicide than women, with similar patterns in other high-income countries [1,2]. High male suicide rates are thought to be influenced by a multitude of factors, including more frequent use of lethal means [3], alongside a tendency for self-reliance [4], which can precipitate hesitancy to reach out for help in times of distress [5].

To date, there has been minimal work unpacking men’s use of, and experiences with, telephone helplines for distress (referred to as helplines for the remainder of this article). Helplines are services that are an integral part of community mental health care systems and are often best placed to interrupt and prevent suicides [6]. Helplines provide immediate and anonymous support for people experiencing distress and/or who are unable to cope with difficulties in their lives [6]. Helplines are staffed by a mix of paid and volunteer professionals or paraprofessionals who are trained in crisis intervention [7,8]. In Australia, helplines offer support for a range of social and emotional problems, including suicide and crisis intervention (e.g., Lifeline, Sydney, Australia), broader mental health support (e.g., Beyond Blue, Melbourne, Australia) and tailored support for specific populations, including men (e.g., Mensline, Footscray, Australia), and specific presenting issues (e.g., sexual assault and family violence; 1800-RESPECT). While not all helplines in Australia are targeted towards crisis or suicide intervention specifically, given links between depression, anxiety, loneliness, relationship breakdown and suicidality for men [9,10,11,12], helplines here are broadly framed as being a vital preventative measure against suicide across all stages of distress and associated precipitating factors.

Calls to helplines in Australia increased significantly during the COVID-19 pandemic. In a 4-week period in 2021, Lifeline saw a 14% and 33% increase on the same periods in 2020 and 2019, respectively [13]. These figures reflect the significant psychosocial toll of COVID-19 related restrictions on Australians [14,15]. The COVID-19 context provides a unique opportunity to explore help-seeking men’s use of helplines at a time where public awareness of mental health and helpline use is at an all-time high. The substantial increase in service use during this time, and the subsequent experiences of users, may have long-term impacts on help-seeking behaviours of men and attitudes towards helpline services post-pandemic.

Helplines overcome a range of barriers to accessing mental health care due to their accessibility, anonymity and affordability [6]. Many men, however, may face additional and unique barriers to engaging with these services. Common attitudinal barriers impacted by masculine norms such as self-reliance and stoicism likely impact the way men present to helplines as the requirements of engaging with helpline support (e.g., emotional disclosure) stand in contrast with these traditional masculine norms [5,16]. Some men may be inclined to minimise symptoms or not disclose distress or suicide risk in an attempt to maintain masculine status [17,18]. Additionally, men may experience and present with unique manifestations of depression [12,19], anxiety [20] and suicidality [21]. The time and information limited context within which helpline interactions occur can add additional complexities in building rapport and connection. Critical aspects of men’s engagement with other forms of mental health support such as establishing trust, transparency, orientation to counselling and shared control over treatment [22] are either less applicable or likely function differently within a time-limited phone call, and without aiding cues such as body language and facial expression. Despite these unique factors, there is currently no evidence base to tell us what men need, expect and value in calls to helplines, nor whether helplines are currently meeting these targets.

An existing body of research highlights the efficacy of helplines in supporting callers to manage distress and crisis [23,24]. Yet information regarding engagement for men in particular is limited due to methodological challenges in gaining demographic information from callers [25]. Further, subjective male caller experiences of these services are under-researched. Expectations of masculine norms and assumptions of gender have been found to impact care in therapeutic settings [22,26], and thus it is practicable that similar implicit biases may impact the type of care received by male callers to helplines. There is some evidence to suggest that helpline counsellors may respond distinctly to male callers, displaying an increased vigilance to implement suicide interventions associated with perceptions of increased suicide risk among men [27]. This propensity for action-oriented intervention styles may result in a caller’s other needs (e.g., need for emotional support) being missed, overlooked or deprioritised. Additionally, evidence suggests that helpline counsellors make decisions about suicide interventions based on recognising patterns of cognitions or emotions presented in a call, and that such patterns may be gendered. For example, helpline counsellors associate engaging in risky activities as a risk factor for male callers, and withdrawal from friends, family or society as a risk factor for female callers [28]. In an analysis of calls to Mensline, Feo & LeCouteur [29] found that helpline counsellors tended to direct male callers to a problem-solving model, even when the man verbalised emotional support and talking as the main reason for call. A few studies have looked at men’s help-seeking on medical helplines, indicating that some men display a reluctance to engage with the service, demonstrated by rationalising their call as being encouraged by someone else and minimising their symptoms or concerns [30,31]. In sum, this body of work suggests that helplines as a service may function uniquely for male callers due to male-specific presentations and needs, and crisis supporter assumptions about the nature of a caller’s distress and necessary corresponding intervention techniques.

In this study, we aimed to (a) characterise men who use helplines as a form of support, (b) explore the impacts of COVID-19 restrictions on men’s helpline use, and (c) provide preliminary insights into men’s experiences with these services. These data will help inform helpline service providers about current levels of service satisfaction among men and illuminate opportunities for optimising helplines with men in mind.

## 2. Materials and Methods

The study is reported according to STROBE guidelines for reporting observational research [32]. Ethical approval for the study was granted by The University of Melbourne Human Ethics Sub-Committee (ethics ID: 2021-13657-22724-4).

### 2.1. Participants and Procedure

Data presented here are a subset of data from a larger online survey in which Australian men aged ≥ 16 years were invited to answer questions about their mental health and any help-seeking experiences during the COVID-19 pandemic, hosted by Qualtrics. In addition to the items presented in this study focusing on helpline use, the larger survey (for self-identified Australian men only) included questions about men’s mental health, social connectedness, resilience, coping, and other help-seeking experiences. While all participants were asked to respond to all items, this study focused on data from questions related to helpline use.

Recruitment occurred from 25 October to 29 December 2021, via targeted Facebook advertisements to Australian men nationwide. These advertisements appeared for participants in the normal advertising panes on Facebook, and they were free to choose whether or not to click through the advertisement. Advertisements contained the following text, adapted from our previous male-specific help-seeking surveys [33]: “*Survey for men: Have you had any difficulties with your mental health during the COVID-19 pandemic? We want to hear about your experience. Complete our short 10–15-min survey here.*”

Participants who clicked through the advertisement link were immediately presented with a plain language statement and consent form with a yes/no response prompt. Consenting participants were then asked to work through the survey, which contained a range of closed and open-ended questions exploring mental health and help-seeking experiences throughout the COVID-19 pandemic. Participants were given the option to enter a draw for a $500 voucher as compensation for their time.

### 2.2. Measures

#### 2.2.1. Suicidality

Three questions were used to assess participants’ suicidality since the beginning of the pandemic (March 2020). These were adapted from Ten to Men, a longitudinal Australian study exploring men’s social and emotional well-being across the lifespan [34]. Questions addressed suicidal ideation, planning, and behaviour: *Since March 2020 (the beginning of the pandemic), have you seriously thought about killing yourself? (Yes/No), Made a plan about how you would kill yourself? (Yes/No), Tried to kill yourself? (Yes/No).* Current suicidality was measured using an item from the Patient Health Questionnaire-9 (PHQ-9) [35]. For the PHQ-9, respondents rate items relative to the preceding two-week period on a scale from 0 “not at all” to 3 “almost every day”. Higher scores indicate higher levels of depression. In this study, any response above 1 (“several days”) to the item: *Thoughts that you would be better off dead, or thoughts of hurting yourself in some way*, was indicative of current suicidality. Cronbach’s alpha in this sample was 0.920.

#### 2.2.2. COVID-19 Stressors

Four questions adapted from Ogrodniczuk et al. [36] were used to assess the psychosocial impact of COVID-19. Two of these were Likert scales: *To what extent has the COVID-19 pandemic put financial stress on you?* (1 = No stress, 5 = Extreme Stress) and *How have government COVID-19 restrictions affected your mental health?* (1 = Very Positive, 5 = Very Negative). Higher mean scores are indicative of higher levels of stress or negative experience. The remaining two questions were categorical: *Have you lost your job due to COVID-19?* (Response categories: No, but hours reduced; No, but job loss is expected; No, but I have had to work from home; Yes, and I have not found another job; Yes, and I have found another job; No, my work has not been affected; No, I do not work at all) and *To what extent has COVID-19 affected your relationship with your partner?* (Response categories: I am not in a relationship; My relationship ended during COVID-19; Our relationship is a lot better; Our relationship is moderately better; No change; Our relationship is moderately worse; Our relationship is a lot worse).

#### 2.2.3. Helpline Use

Given the paucity of research in the area, bespoke items were generated for this study to assess participants’ use of, and experiences with, helplines throughout the pandemic. Firstly, participants were presented with the following item: *Have you contacted any of the following mental health support/crisis lines since March 2020 (i.e., the beginning of the COVID19 pandemic)? Tick all that apply.* A list of nine Australian helplines for distress were provided, along with a free-text entry option for participants to specify a service not captured in the list. If any helplines were ticked, participants were presented with the following free-text entry item: *Please comment on your experience with these helplines/support services (e.g., your satisfaction with the service/support)*. Participants were asked to provide one qualitative response regardless of the number of helplines they had contacted, as the purpose of the study was to obtain feedback on helplines in general rather than service specific feedback.

### 2.3. Data Analysis

#### 2.3.1. Quantitative Analysis

All variables were examined descriptively. The content of responses to “other” fields for demographic items was reviewed collaboratively by three authors (KT, MW and ZS) and assigned to existing categories or new categories created based on content. Based on responses to the question *Have you contacted any of the following mental health support/crisis lines since March 2020 (i.e., the beginning of the COVID19 pandemic)?* a binary yes/no variable was created for helpline use. For the categorical question *Have you lost your job due to COVID-19?* responses were simplified to compare those experiencing job loss/a reduction in work hours and those not experiencing job loss. Chi-square and t-test analyses were used to evaluate differences in the demographic profile and COVID-19 stressors of men who did and did not use helplines. For all chi-square analyses, Fisher’s exact test was applied where expected cell counts were ≤5 and effect sizes were examined according to Cramer’s V (φc), where 0.1 is considered a small effect, 0.3 a medium effect and 0.5 a large effect. For *t*-tests, effect sizes were examined using Hedges *g* where 0.2 reflects a small effect, 0.5 a medium effect and 0.8 a large effect. Post hoc testing using adjusted residuals with Bonferroni corrected p values were used to further examine between group differences in chi squares. All quantitative analysis was completed in SPSS 27.

#### 2.3.2. Qualitative Analysis

Responses to the open-ended question: *Please comment on your experience with these helplines/support services (e.g., your satisfaction with the service/support)* were analysed using inductive thematic analysis involving a six-stage process of coding and theme development [37,38]. First, all responses were read and re-read in detail to gain familiarity with the data. Then, responses were downloaded into a spreadsheet in preparation for analysis. Responses were then independently coded by the first author (KT), and codes were developed to encompass similar responses and labelled descriptively. Codes were then organised under higher order categories via research consultation between three authors (KT, MW and ZS), with any disagreements discussed until consensus reached. For example, the codes ‘*calming down and getting through immediate crisis*’ and ‘*felt heard*’ were subsumed into the subtheme ‘*identifying and responding to caller needs*’. At this stage, categories were grouped and subsumed resulting in three broad themes: *positive caller experiences, negative caller experiences*, and *mismatched expectations*. This initial thematic structure was then reviewed by a fourth author (SR), which resulted in a structural shift of themes to better demonstrate the underlying thematic content of responses rather than response valence (positive versus negative). As such, codes and categories were reorganised into the present themes and reviewed by all authors. Finally, consensus of theme names, descriptions and illustrative quotes were made through author meetings and writing of the manuscript. Names of specific helplines mentioned in responses have been removed.

## 3. Results

### 3.1. Quantitative Results

#### 3.1.1. Sample Characteristics

In total, 812 men completed the survey. Of these, 684 (84.2%) provided responses related to helpline use and are therefore included in this study. Within the sample, 100 (14.6%) of the men selected responded that they had used one or more helpline telephone service since March 2020, and 584 (85.4%) had not used a helpline service. Men were aged between 17–83 years (*M* = 50.13, *SD* = 15.20). The majority of the sample comprised men living in metropolitan areas (65.9%, *n* = 451), who were partnered or married (61.5%, *n* = 421), and self-identified as heterosexual (70.5%, *n* = 482). Participant demographics and mental health measures are presented in Table 1, separated by helpline use. Age differences were present between the groups, with an increased number of participants in between 18–25 years using helplines, AR = 3.00, *p* < 0.001. There was a significant difference in the employment status dependent on helpline use, with post hoc residual testing revealing that those men using helplines were significantly more likely to be unemployed, AR = 3.20, *p* < 0.001.

Lifeline was the most utilised helpline service, with 44% (*n* = 44) of participants who had used a helpline selecting Lifeline, followed by Beyond Blue (42%, *n* = 42), Mensline (29%, *n* = 29) and e-headspace (12%, *n* = 12). See Table 2 for a complete list of helplines used by participants.

#### 3.1.2. Impact of COVID-19 Stressors

Men who used helplines were significantly more likely to report having suffered financial stress due to the pandemic (*M* = 2.99, *SD* = 1.51) than those who did not use helplines (*M* = 2.13, *SD* = 1.28, *t*(106.05) = 4.97, *p* < 0.001, g = 0.65), and to have experienced either job loss or a reduction in employment hours than those who did not use helplines, χ^2^(1, 471) = 8.51, *p* = 0.004, φ_c_ = 0.13. They were also significantly more likely to report that pandemic restrictions had a negative impact on their mental health (*M* = 4.06, *SD* = 1.04) than those who did not use helplines (*M* = 3.73, *SD* = 0.96), *t*(607) = 2.91, *p* < 0.001, g = 0.34). Further, there were significant differences in the perceived impact of COVID-19 on the participants’ intimate relationships between the two groups, χ^2^(3, 341) = 18.28, *p* < 0.001, φ_c_ = 0.21. Post hoc residual testing revealed that those who used helplines were more likely to have experienced a relationship breakdown during the pandemic, AR = 3.67, *p* < 0.001. No significant difference was found between groups in terms of the impact of the pandemic on perceived quality of their relationship.

### 3.2. Qualitative Results

Ninety-two participants who had used helplines provided qualitative information on their experiences. Three broad themes were developed from the data, recounting men’s reports of positive and negative aspects of the services they accessed. The first theme, *Nature of interaction,* details the aspects of an interaction between caller and counsellor that influence men’s attitudes towards and experiences with helplines. The second theme, *Structure of services,* articulates how men use helplines within the wider mental health system, and how structural elements of helplines have the potential to both help and harm callers. Finally, the third theme, *Mismatched expectations,* describes men’s perceptions of the limitations of helpline services and a possible mismatch of expectations regarding the marketing of helpline services and what they can offer. Though the themes are presented as discrete, some participants gave responses that fell across themes, highlighting the variation in quality of experience across different services and different points in time. As such, the themes can be viewed as interconnected.

#### 3.2.1. Nature of Interaction

This broad theme centred on the connection between caller and counsellor as key in determining the quality of interaction and level of satisfaction felt by participants. Through their responses, participants indicated that a counsellor’s ability to build rapport, be attuned to their needs and respond accordingly could prove influential in callers’ satisfaction with their experience. Firstly, this theme details how individual characteristics and behaviours of the counsellor shaped men’s experience with a service, and therefore impacted their attitudes towards helplines. Secondly, it sheds light on men’s diverse needs when contacting helplines, the importance of counsellors in recognising that need and the potential for adverse consequences when needs are not recognised.

Some participants spoke highly of the quality of the counsellors they connected with, reporting feeling heard, engaged and cared for in their interactions. Participants described their telephone counsellors as supportive, friendly and skilled, speaking to the importance of rapport building and connection even within such fleeting encounters for men seeking support:


*I spoke with someone who was kind and supportive and helped me see a way out of a difficult situation.*



*(Helpline) lady was really nice, helped me unbottle some of my emotions…*


By contrast, some men reported difficulty in connecting with their counsellors and the negative impact this had on their experience. While these particularly unfavourable accounts were few in number, they are important to highlight as they indicate a varying standard of service delivery across helplines:


*No help whatsoever, the person at the end of the line was useless.*



*The service provider agreed that one of the reasons I was calling was ‘terrible’ but after I ‘rambled’ she broke in with ‘I’m not able to continue this call. What are you going to do when this call ends? I stated that I would either ‘do dishes’ or ‘go for a walk’.*


It is evident that these experiences had negative impacts on the caller, resulting in increased distress at a time when they were vulnerable.

Moving beyond rapport building, participant responses gave preliminary insights into the needs of male callers when contacting helplines, and counsellors’ ability to connect with and respond to those needs. Some participants spoke of counsellors’ ability to de-escalate their emotional distress or level of arousal, commenting that their interactions were helpful in calming them, reassuring them or getting them through a crisis:


*They were outstanding. Spent plenty of time with me and really helped calm me and reset.*



*Useful to get me out of a bad space. Nice to simply have someone to share my thoughts with and listen without judgement.*


Included in the above, a counsellor’s ability to simply listen non-judgmentally seemed vital to some men’s satisfaction with services. These responses suggest that for some men, all that was required to alleviate distress was the opportunity to tell their story and feel heard:


*They listened, but I found they just provided an ear. Which is important in the situation. I called when I was at my lowest.*


Conversely, some men did seem to appreciate advice, tips, or problem-solving strategies given by some helpline counsellors:


*It was very helpful with some tips to better manage what I was going through.*


These responses suggest men contact helpline services wanting diverse forms of support from their counsellors, and that there is a need for counsellors to recognise and respond to this diversity. For example, the participant quoted below seemed disappointed that the helpline counsellor was more concerned about their suicidal thoughts than about the situational factors that were contributing to their distress. This demonstrates that although managing risk by focusing on current suicidality is essential in ensuring caller safety, doing so at the cost of focusing on other concerns raised may lead to men not feeling heard or validated by the service, and ultimately limit the opportunity for a reduction of distress:


*Found (Helpline) uninterested in my problems concerning domestic abuse by an ex-girlfriend, as I was a male victim. Only concerned as to my suicidal thoughts. Useless for assistance to male victims of DV/DA. Would never recommend to male friends. Gave the appearance of bias against males.*


Further, this response highlights additional difficulties faced by men who are victims of domestic abuse and how a counsellor’s response may serve to reinforce the stigma male victims experience when attempting to seek help for their concerns. The participant’s reaction to ‘*never recommend (the helpline) to male friends*’ demonstrates the consequences that services that are perceived as ineffective can have on the culture of help-seeking behaviours and role modelling among males more broadly.

In sum, this theme highlights the aspects of an interaction that were pivotal in driving the level of (dis)satisfaction felt by male callers. Through references to genuine connection, building rapport and responding to needs, responses here demonstrate what men did and did not value within a helpline interaction, and the counsellor traits, skills and behaviours that impacted men’s experiences with helpline services both positively and negatively.

#### 3.2.2. Service Structure

The second theme relates to the structural elements of helpline services that shape their unique role in the mental health care system and highlights helpful and harmful aspects of helplines viewed through the eyes of men. On the one hand, responses in this theme highlighted the ways in which the structure of helplines (free, immediate and accessible services) allows them to provide a valuable and useful service to men in distress. On the other hand, the structure of service delivery and technological limitations meant that some men experienced incomplete or interrupted calls where they were unable to get through to a helpline counsellor, resulting in increased distress.

Participants commented on the structure of services in facilitating a positive interaction, citing speed, accessibility and connection to other mental health services as key reasons for using helplines.


*Useful to deal with immediate overwhelming thoughts.*


Participants commented on the role of helplines as a bridge to accessing face-to-face mental health services, or as an adjunct service to supplement ongoing clinical care when support is urgently required:


*Was good to talk to someone immediately as opposed to waiting for my next session with (my) psychologist.*



*It was instant, connected me to a counsellor right away, they really listened and referred me to someone in my area who I can see face-to-face once lockdown ends.*


The above participant also referenced lockdowns due to the COVID-19 pandemic and the difficulty in accessing face-to-face care in this time. This highlights the unique purpose of helplines over the last two years in supporting men during the pandemic when distress may have been increased, and regular services may have been harder to access.

A few participants reported that their contact with a helpline directly interrupted a suicide attempt, reaffirming the role of crisis helplines as a life-saving service. The participant quoted below affirmed the necessity of immediate services for suicidal men, as he noted that the ability to reach out to someone at the exact time that he was experiencing suicidal thoughts was instrumental in his positive account of the services, which likely contributed to saving his life:


*(Helpline) was great. The counsellor was brilliant. I was thinking about throwing myself in front of a train and called. He was the best I’ve ever spoken to.*


Despite being applauded for their accessibility, many men lamented the difficulty in getting through to speak to counsellors. These participants spoke of long hold times, being hung up on or calls being left unanswered:


*I was on hold for 45 min and then had to go back to my kids without speaking to anyone. It was very disappointing.*


The impact of not being able to access help when in a state of distress was clear, with some reporting that they *‘gave up waiting’*. This sense of disappointment may have consequences on men’s likelihood of seeking help from a similar or more formal service in the future. Further, one participant noted that although they got through to speak to a counsellor, the call was terminated, leaving them more distressed than prior to making the call and exposing the technological limitations of telephone services:


*(Helpline) were useless and hung up on me leaving me upset, bewildered, anxious and feeling worthless and more alone than ever.*


Similarly, the participant responsible for the quotation below reasoned that overwhelmed services understandably impacted the quality of care he received:


*There didn’t seem to be much they could do for me as they were so overwhelmed, but pointed me in other directions.*


This theme demonstrates that men value helplines as a form of support for their accessibility and immediacy, both vital structural elements of helpline service provision. Further, it provides evidence that men may use helplines both as a stand-alone service and as an adjunctive to long-term clinical care. It also highlights that this model of service delivery—where instant support is expected—may inadvertently cause harm when overwhelmed services and technological limitations result in calls going unanswered or cut short.

#### 3.2.3. Mismatched Expectations

The final theme focuses on responses that exposed the need for setting realistic and pragmatic expectations regarding the type of support that helplines can feasibly offer.

Some men spoke about their dissatisfaction in terms of services not being able to address their complex psychosocial issues within the short time frame of service provision, or expressed a preference for in-person services to achieve long-term positive outcomes:


*Important and useful in the moment. Ultimately not a satisfying service to reach outcomes.*


This response, like others, demonstrates how some men were cognisant of the practical limitations of helplines, recognising their value in providing immediate support while still noting feeling unsatisfied or that the service was unable to meet all of their needs.

Others seemed less sympathetic to the constraints of these services, placing responsibility on the helplines for not being able to provide resolutions for their complex situations and stressors causing them distress:


*To seek help and how could I overcome these problems, I contacted (Helpline) but harassment from ex-wife and her father is too much that calling on (Helpline) didn’t help me. I’m still very stressed.*


Further, some men expressed doubt at the ability of helplines to provide adequate care due to counsellors not being qualified mental health professionals:


*Waste of time because not a qualified psychologist.*



*I am also dubious about the qualifications of the counsellors.*


Threaded throughout these responses is a sense that some men may approach helplines with the notion that the service will not be able to help them either based on preconceived ideas or past experience, resulting in increased barriers to a positive interaction from the outset:


*At the end of the day they can’t fix your problems, it just brings them to the surface. I just bottle everything up. I don’t feel like anyone can help.*


Lastly, some men indicated that the format of helplines did not suit them, citing a personal preference for other forms of support:


*Over the phone has always been a bit weird, I never really vibe it.*



*I much prefer talking to a friend if one is available.*


Taken together, this theme recognises that telephone helpline services may not be suitable or effective for all men as there is wide variation in personal preference, presenting concerns and caller needs. While some callers realistically acknowledged the inability of a short-term helpline service to meet their often-complex needs, reflections from many participants signalled a clear mismatch between expectations of a helpline service and outcomes achieved in the immediate term.

## 4. Discussion

This study examined helpline use among a sample of Australian men during the COVID-19 pandemic. Findings presented here provide a preliminary snapshot of the profile of men who use helplines, what their needs are and their current levels of service satisfaction. Men in the sample displayed similar demographic profiles regardless of helpline use, with the exception that men using helplines were slightly younger and more likely to be unemployed. Lifeline was the most utilised helpline by men in the sample (44%), followed closely by Beyond Blue (42%) and Mensline (29%). Men who used helplines were more likely to have experienced suicidality (intent, plan, and attempt) since the onset of the pandemic than men not using helplines. They were also significantly more likely to have also sought help from a mental health professional, indicating that men using helplines were seeking multiple forms of support to manage their distress. Our findings also provide insights into the experience of helpline use and level of service satisfaction felt by men. In sum, the qualitative findings presented here add to the growing body of evidence that positions men as a diverse group of help-seekers [29,39] who use helplines as a form of support in a variety of ways. Responses also provide insights into a level of service (dis)satisfaction felt by some men, impacted by elements of the interaction, the structure of services, and the level of knowledge or expectations about services that men arrive with.

Men using helplines were significantly more likely to report being negatively impacted by stressors induced by the COVID-19 pandemic, reporting increased financial stress and work insecurity, negative impacts on mental health and relationship breakdown. Our findings indicate that men in our sample experiencing these stressors in Australia due to COVID-19 were seeking support through available and accessible means such as helplines. This is an important and positive finding, given strong links between financial strain and relationship breakdown and suicidal behaviour in men [9,36,40]. Linking this with an overall increase in helpline service use during this time [13], these findings may point to the benefit of increased public discourse around mental health during the pandemic, potentially norming help-seeking behaviours for men. Further, given high service demand [13] exacerbated by the pandemic, helplines may have been a viable accessible option for men needing help with these situational stressors.

When asked about their experiences with helplines, many men spoke about the factors relating to the nature and quality of the interaction they had with their helpline counsellor. Men cited connection with their counsellor as a key component of service provision, valuing counsellors who displayed kindness, support and care. This finding contributes to the literature around men’s needs and preferences when using helplines, as they alluded to a diverse range of needs including de-escalation and crisis intervention, listening and support and advice or access to further referrals. This largely aligns with recent findings that the most common expected outcomes of calling Lifeline are to ‘feel heard and listened to’, followed by to ‘receive safety advice or support to stay safe’, with no differences in expected outcomes between gender groups [41]. Challenges with establishing a therapeutic connection with male clients has been cited by therapists previously [26], in particular with therapists reflecting a sense that if they do not secure a rapport within the first session, many men will not return to give them a second chance. Given the comparably short time frame helpline counsellors have to connect and engage with their callers compared with therapists, the potential for increased difficulties in connecting with male callers is high and is likely to have significant impacts on caller outcomes, warranting future research attention.

Findings here point to the necessity for helpline counsellors to become attuned to an individual callers’ needs through either picking up on cues or seeking their preference in that moment (i.e., emotion focused vs. problem focused support), rather than relying on assumptions of what men want [22]. Having said that, recognition of the challenges that helpline counsellors experience in quickly identifying needs and applying correct judgments and responses tailored to men utilising helpline services without access to non-verbal cues is required [42]. As such, further research is needed to identify which men are likely to prefer emotional support relative to proactive problem solving, and whether their preferences may differ depending on their varying circumstances. This will further our understanding of how helpline counsellors can best identify needs quickly and meet those needs accordingly.

An important finding here was that for some men experiencing suicidality, their counsellor’s attempt to manage suicide risk resulted in their other needs being missed, leaving them feeling unhelped. Asking about suicide and attending to suicidal thoughts are key requirements of support in many helplines, as they should be. However, some suicidal men may require more holistic support in that moment to alleviate distress, rather than just crisis assessment and intervention. While this study does not offer a comparison to other gender groups’ experiences, this finding complements the observation that helpline counsellors are more likely to employ suicide prevention interventions when suicide risk is identified in men compared to women [27] and provides insight from men that such an approach may interrupt connection, lowering engagement on the call.

Findings indicate that structural aspects of helpline service delivery impacted men’s experiences. Supporting much previous literature, men in our sample valued helplines for their immediacy and accessibility as a support service for managing acute distress [6,42]. Our qualitative findings indicate that men use helplines in a variety of ways, both as stand-alone support in times of acute crisis and as an adjunct to more long-term psychological care. This is supported by the quantitative finding that 84% of those who used helplines in the sample reported that they had also been in touch with a mental health professional since March 2020. Coveney and colleagues [43] report that one third of their sample of people using helplines reported engagement with other mental health professionals, indicating that callers contact helplines in specific times of acute need or crisis, adjunctive to psychotherapy support. Our findings confirm the critical importance of helplines for men in intervening during suicide attempts, providing emergency support and assistance at times when men either cannot, or choose not to, access other forms of support. This is critically important given increasing demand on mental health services across the country, exacerbated by the COVID-19 pandemic [13]. Ensuring helplines offer consistent and high-quality support service to men is also key given that helplines can act as the bridge to accessing further mental health services for men, indicated by those who spoke of referrals. Given what we know about men’s increased barriers in entering and engaging with the health care system [22,44], effective and accessible entry points are vital.

The praise for immediacy and accessibility of services was in tension with some reports of long wait times and challenges in getting help. Within men’s responses, it was clear that being left on hold or disconnected was disappointing, and either sustained or intensified their level of distress. Lifeline, Australia’s largest provider and the most utilised helpline choice within the sample, reported an average wait time of 70 s and a call answer rate of 90% in 2021, indicating that these experiences are, in general, an anomaly [45]. Having said that, the reality of service-capacity limits may reinforce negative attitudes towards mental health services and help-seeking for some men. In light of evidence from psychotherapy research that shows dissatisfaction with therapy can deter men from future engagement with services and, more broadly, disclosure of distress [46], attention to, and funding of, enhanced helpline capacity is essential. Future in-depth research to better understand the specific needs of male helpline callers will assist in tailoring service responses.

Some participants’ comments reflected a limited understanding about the type and level of support that helplines can feasibly offer. While some participants recognised the limitations of helplines as a one-off support service, others seemed more disgruntled with the service due to a perceived inability to tackle complex situational stressors or mental health conditions within the span of a phone call. While helplines are an invaluable and needed resource, previous evidence nonetheless suggests that impacts may be relatively small and short-term in nature [23]. Research also suggests that some community expectations around helplines in general can be misguided, with two thirds of participants surveyed about Lifeline indicating they would expect Lifeline to be involved in their long-term care plan [41]. This suggests that population knowledge of services is not in line with actual service delivery realities and constraints, and this seemed to be the case for many men in our sample. Our findings mirror previous work within the therapeutic context, where therapists cited that men often wanted a ‘quick fix’ for complex issues [26]. We reaffirm and extend the argument of Seidler and colleagues [26] that more work needs to be done in setting realistic expectations for men and providing education about what to expect upon contact with services, specifically with regard to the context of helpline services. Reiterating the purpose and scope of services to men may help to ensure expectations of services are in line with the reality of what they are able to offer. Further, as evidenced in participant responses, a sense of disappointment was felt by men when help-seeking efforts were viewed as futile, and problems seem unfixable by available support services. Appropriate orientation to treatment and expectation setting is a key facet of engagement in psychotherapy among men [47]. As helpline counsellors are unlikely to have time to clarify the purpose of the service at the time of a call, there is scope for public mental health promotion messaging to better clarify what helplines can feasibly help with. This may aid in ensuring that male callers’ expectations are realistic on presentation to helpline services and in increasing their confidence and trust in the service.

Some men voiced their doubt about the qualifications or skill level of helpline counsellors. For most helplines, the paraprofessional and/or volunteer status of most helpline counsellors is in fact a key element of the service that allows them to function within the scope and reach that they do. The expectation of being able to talk to a registered mental health professional may also represent a lack of knowledge in terms of the role of these services within the broader mental health care system. The belief that volunteer staff will be unable to provide adequate support is also interesting given some evidence that suggests volunteers display more empathy and often achieve better outcomes in helpline calls compared to paid professionals with a mental health or counselling qualification [48]. Importantly, helpline workers in Australia are trained in specific forms of crisis intervention tailored to the helpline context. For example, Lifeline is a Registered Training Organisation (RTO) whose helpline counsellors are trained in the nationally accredited Crisis Supporter Workplace Training (CSWT) [49]. Regardless of the qualification of helpline counsellors at any given service, it is important that men who do call services are confident that the person they speak to is able to provide the necessary support in the moment, and that men are knowledgeable about the type of support that can be expected within a helpline context.

### 4.1. Implications

The present study reinforces the value of helpline services for Australian men and the importance of ensuring these services remain accessible and affordable. The findings also reinforce the notion that support for men does not fit into a one size fits all approach, and that a network of well-funded and connected services with clear referral pathways will help to ensure that men are able to access support that works for them when they need it. The findings have several practical implications. First, the findings underscore a need for enhanced training for helpline counsellors around male-specific presentations of distress and how to effectively engage with and support male callers, as is underway in the therapy space [50,51]. Any such efforts should be underpinned by evidence and involve co-design elements with men who have a lived experience of helpline use to maximise their impact potential. Second, findings highlight the need for ongoing and sufficient funding of helplines to ensure that services are able to provide a high level of care at the right time for all callers. Services would benefit from enhanced mechanisms for evaluation, whether through ongoing monitoring of quality of care, or increased opportunities for service users to provide feedback. Finally, while participants who used helplines displayed awareness of the services, there is an opportunity for helplines to invest in public awareness campaigns and marketing strategies that extend beyond providing awareness that the services exist, but rather outline the precise structure, purpose, and scope of services, and provide realistic information on what a caller can expect when calling a specific helpline in comparison to other services. It would be beneficial to consider how future awareness campaigns may specifically engage with and target men, for example through partnerships with male dominated work and recreation spaces, and utilising male friendly language and imagery [52,53].

### 4.2. Limitations and Future Directions

Our study had several limitations. Firstly, although the approach of gathering qualitative data through an open-ended survey question is an established one [54] and allowed for a substantial number of qualitative responses and provided insights into a diverse range of experiences, it did not provide the scope to probe for further depth or elaboration of responses. This limited our ability to discern participants’ thoughts or experiences outside of their verbatim responses. Second, responses were general and provided no detail around specific services or instances of service use. Many participants selected multiple helplines, but provided a broad response about their experience, limiting our ability to draw conclusions about specific services. Further, while the data were collected during the COVID-19 pandemic, the specific impact of the pandemic on men’s experiences of helplines is unclear beyond the evidence that those using helplines were more likely to have experienced COVID-related stressors. As such, we cannot be sure whether responses would have been different outside of the pandemic context. Finally, the sample is not representative, and respondents likely represent those inclined both to seek help and to speak about their experiences, meaning findings may not generalise to the wider population. Notwithstanding this, the present data provide a valuable snapshot of helpline usage among Australian men that provides important groundwork for substantiation and within-service research in the future. Further qualitative inquiry involving in-depth interviews with men about their experiences of helplines are required to ascertain a deeper understanding of the needs, preferences and experiences of men using these services. Further, the view of helpline counsellors and their supervisors is not captured in this research and would help to strengthen the evidence base by providing insights into what counsellors find challenging about supporting men within this context. Taken together, such research would provide a base on which to develop ways to optimise helplines to meet the needs of male callers, a need that has been previously identified [42].

## 5. Conclusions

This study extends our understanding of men as help-seekers by identifying subjective experiences of utilising Australian helplines. While many men valued helpline services, it is clear that elements of engagement and service provision could be enhanced to better meet the needs of Australian men. Further in-depth research is required to extend these findings and identify opportunities to tailor helpline services for men based on their experiences and needs. Through this, we can better equip helpline counsellors with the knowledge, skills and confidence to connect with and support male callers to helplines.

## Figures and Tables

**Table 1 ijerph-19-09143-t001:** Participant demographics and key variables by helpline use.

	Total (*N* = 684)	Helpline Use (*n* = 100)	No Helpline Use (*n* = 584)	*t*/Chi (df)	Sig	ES (g/φ_c_)
**Mean age (SD)**	50.13 (15.20)	46.65 (15.29)	50.73 (15.12)	**2.49 (682)**	**0.013**	**0.269**
	% (*n*)	% (*n*)	% (*n*)			
**Age groups**				**12.27 (3)**	**0.007**	**0.134**
18–25	6.9 (47)	14.0 (14)	5.7 (33)			
26–45	31.3 (214)	30.0 (30)	31.5 (184)			
46–60	34.1 (233)	37.0 (37)	33.6 (196)			
61+	27.8 (190)	19.0 (19)	29.3 (171)			
**Sexuality**				1.16 (1)	0.282	0.041
Heterosexual	70.5 (482)	75.0 (75)	69.7 (407)			
Sexual minority	29.5 (202)	25.0 (25)	30.3 (177)			
**Relationship status**				2.82 (1)	0.093	0.064
Single/never married	38.5 (263)	46.0 (46)	37.2 (217)			
Married/Partnered	61.5 (421)	54.0 (54)	62.8 (367)			
**Residence**				1.05 (2)	0.591	0.039
Metropolitan	65.9 (451)	68.0 (68)	65.6 (383)			
Regional	27.8 (190)	28.0 (28)	27.7 (162)			
Remote or rural	6.3 (43)	4.0 (4)	6.7 (39)			
**Employment status**				**14.06 (3)**	**0.003**	**0.143**
Employed	65.8 (450)	62.0 (62)	66.4 (388)			
Unemployed	10.1 (69)	19.0 (19)	8.6 (50)			
Retired	20.3 (139)	13.0 (13)	21.6 (126)			
Student	3.8 (26)	6.0 (6)	3.4 (20)			
**Education**				0.95 (1)	0.329	0.037
High school/Trade/cert/diploma	42.5 (291)	47.0 (47)	41.8 (244)			
Undergraduate/postgraduate degree	57.5 (393)	53.0 (53)	58.2 (340)			
**Income**				4.62 (4)	328	0.082
0–49,000	36.0 (246)	44.0 (44)	34.6 (202)			
50,000–99,999	29.8 (204)	30.0 (30)	29.8 (174)			
100,000–149,999	19.4 (133)	15.0 (15)	20.2 (118)			
150,000–199,999	6.4 (44)	4.0 (4)	6.8 (40)			
200,000+	8.3 (57)	7.0 (7)	8.6 (50)			
**Have sought help from mental health professional since March 2020**				**55.10 (1)**	**<0.001**	**0.284**
Yes	49.7 (340)	84.0 (84)	43.8 (256)			
No	50.3 (344)	16.0 (16)	56.2 (328)			
**First time help-seeking since March 2020**	***n* = 340**	***n* = 84**	***n* = 256**	0.92 (1)	0.338	0.052
**Yes**	23.5 (80)	27.4 (23)	22.3 (57)			
**No**	76.5 (260)	72.9 (62)	77.7 (199)			
**Suicidality (Since March 2020)**	***n* = 684**	***n* = 100**	***n* = 584**			
Intent	28.4 (201)	56.0 (56)	24.8 (145)	**39.98 (1)**	**<0.001**	**0.242**
Plan	17.7 (121)	41.0 (41)	13.7 (80)	**43.71 (1)**	**<0.001**	**2.53**
Attempt	2.2 (15)	5.0 (5)	1.7 (10)	**4.30 (1)**	**0.038**	**0.079**
**Current suicidality (PHQ-9 item)**	***n* = 684**	***n* = 100**	***n* = 584**	**32.80 (1)**	**<0.001**	**0.219**
Yes	31.4 (215)	56.0 (56)	27.2 (159)			
No	68.6 (469)	44.0 (44)	72.8 (425)			

**Table 2 ijerph-19-09143-t002:** Helpline service usage.

Helpline	% of Participants Used (Multiple Responses Per Participant)
Lifeline	44.0 (44)
Beyond Blue	42.0 (42)
Mensline	29.0 (29)
eheadspace	12.0 (12)
Suicide Call Back Service	3.0 (3)
Open Arms	3.0 (3)
QLife	3.0 (3)
1800 Respect	6.0 (6)
Kids Helpline	2.0 (2)
**Helplines derived from “other” responses**	7.0 (7)
Family Relationship Advice Line	1.0 (1)
Resolve Warm Line	1.0 (1)
Soldier On	1.0 (1)
MATES line	1.0 (1)
Gambler’s Help	1.0 (1)
Other miscellaneous helpline (name not provided)	2.0 (2)

## Data Availability

Data from this study are unavailable due to ethical restrictions.

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
