# Peer review of "“I Called When I Was at My Lowest”: Australian Men’s Experiences of Crisis Helplines"

_ijerph, 2022, doi:10.3390/ijerph19159143_

Round 1

Reviewer 1 Report

Authors have conducted a cross-sectional study analyzing both quantitative and qualitative data about men's experience of helplines during COVID pandemic in Australia.

Introduction is well written, highlighting the importance of helplines, specific barriers for men and provided a good foundation  for the article and discussion section.

Methods section is robust, clearly describing type of analysis and following STROBE guidelines. They have also clearly mentioned how data was collected and qualitative data use categorized using thematic approach. 

Results are well written, and easy to understand for the readers and several interesting outcomes are noticed by the authors which have been utilized in discussion section.

Discussion and conclusion is very robust with discussion of previous studies, their current analysis and the barriers that are anticipated in men population are further highlighted. They were also able to discuss potential changes in the approach to subset of men population based on their qualitative analyses. They have included limitations which are appropriate and mention the implication of their study to potentially tailor the approach by helplines for men population.

In line 148 and 168 the authors need to use initials of authors who reviewed the data instead of writing XX.

Reviewer 2 Report

This manuscript presents results of an online survey of 684 Australian men (100 of whom had called a crisis helpline) to investigate the characteristics and experiences of those who called a helpline compared to those who had not. The survey took place during the COVID-19 pandemic period. Several differences were observed between those who called crisis helplines and those who did not and identified a number of issues regarding the experiences among those who called helplines in this both quantitative and qualitative study.

The greatest strength of the manuscript and study lies in the investigation of a group that has essentially been ignored as a group with regard to crisis helplines, that is, men. The high risk among men for suicide and suicidal behaviors, coupled with evidence that men nonetheless are less likely to seek help from crisis helplines to deal with their traumas and crises, makes this investigation’s information useful and of interest.  Another strength is the identification of the experience men had who called helplines, both positive and negative in nature. Issues that have practical implications for the approach and characteristics of helplines so that men find them as a helping resource in their trauma and crisis are provided. The heterogeneity of men’s experiences with helplines – some positive and some negative – is also important to show that helplines can be a resource for men but recognizing that more could be done to make them more hospitable and engaging to even more men. An additional strength is the identification of numerous themes and issues about helplines and men that are unknown and require additional research so that helplines can be more engaging of men and their needs in times of crises. Finally, the inclusion of qualitative findings enrich the usefulness and implications of the study.

There are some issues and questions that might be addressed.

(1) Details about the survey – In order to replicate this investigation in other locations/countries and also for clarity of description, there are details missing about the survey and its processes.  This survey does not claim to be a random sample, but it also does not provide information about the Facebook advertisements through which the survey recruited participants. Exactly how or where did these advertisements appear? What exactly was the approach of these earlier male-specific help-seeking surveys? This is vital to the sample procedures and at least some basic information should appear here so that readers do not have to seek another source to determine how men had a chance to see and respond to the recruitment request. It is also indicated that these data are from a larger survey. What is the nature of that larger survey? Was it also only recruiting men or is it a broader survey? Is the issue of helplines also the focus of that survey? Did the men in the current data also respond to other questions if the survey subset here is not the entire focus of the other larger survey? The information not provided here need not be in great detail, but some basic understanding of the methodology and possible limitations regarding the sample included seem warranted.

(2) Measures –As both a quantitative as well as qualitative study, characteristics of the measures utilized (e.g., reliability, validity, sources) seem warranted. The PHQ-9 is noted for instance without reference or explanation of the meaning of “PHQ.”

(3) Limitations – A number of appropriate limitations are provided by the authors. However, the issue of the sample and whether it represents a representative sample is not mentioned. This is particularly true with the limited information provided for recruitment processes noted already. A representative sample is not necessary at this point in knowledge on this essentially unstudied phenomenon. 
